# Differentiable Spline Approximations

**Minsu Cho**[1][†]  **Aditya Balu**[2][†]  **Ameya Joshi**[1]  **Anjana Deva Prasad**[2]

**Biswajit Khara**[2]  **Soumik Sarkar**[2]  **Baskar Ganapathysubramanian**[2]

**Adarsh Krishnamurthy**[2]  **Chinmay Hegde**[1]

**New York University**[1], **Iowa State University**[2]
{mc8065, ameya.joshi, chinmay.h}@nyu.edu
{baditya, anjana, bkhara, soumiks, baskarg, adarsh}@iastate.edu [*]

## Abstract

The paradigm of differentiable programming has significantly enhanced the scope of machine learning via the judicious use of gradient-based optimization. However, standard differentiable programming methods (such as autodiff) typically require the machine learning models to be differentiable, limiting their applicability. Our goal in this paper is to use a new, principled approach to extend gradient-based optimization to functions well modeled by splines, which encompass a large family of piecewise polynomial models. We derive the form of the (weak) Jacobian of such functions and show that it exhibits a block-sparse structure that can be computed implicitly and efficiently. Overall, we show that leveraging this redesigned Jacobian in the form of a differentiable "layer" in predictive models leads to improved performance in diverse applications such as image segmentation, 3D point cloud reconstruction, and finite element analysis. We also open-source the code at https://github.com/idealab-isu/DSA.

## 1 Introduction

**Motivation:** Differentiable programming has been a paradigm shift in algorithm design. The main idea is to leverage gradient-based optimization to optimize the parameters of the algorithm, allowing for end-to-end trainable systems (such as deep neural networks) to exploit structure in data and achieve better performance. This approach has found use in a large variety of applications such as scientific computing [Innes, 2020; Innes et al., 2019; Schafer et al., 2020], image processing [Li et al., 2018a], physics engines [Degrave et al., 2017], computational simulations [Alnæs et al., 2015], and graphics [Li et al., 2018b; Chen et al., 2019]. One way to leverage differentiable programming modules is to encode additional structural priors as "layers" in a larger machine learning model. Inherent structural constraints such as monotonicity, or piecewise constancy, are particularly prevalent in applications such as physics simulations, graphics rendering, and network engineering. In such applications, it may be beneficial to build models that obey such priors by design.

**Challenges:** For differentiable programming to work, all layers within the model must admit simple gradient calculations; however, this poses a major limitation in many settings. For example, consider computer graphics applications for rendering 3D objects [Kindlmann et al., 2003; Gross et al., 1995; Loop and Blinn, 2006]. A common primitive in such cases is a *spline* (or a piecewise polynomial) function which either exactly or approximately interpolates between a discrete set of points to produce a continuous shape or surface. Similar spline (or other piecewise polynomial) approximations arise in

---

[*][†]Equal contribution.

35th Conference on Neural Information Processing Systems (NeurIPS 2021).

partial differential equation (PDE) solvers [Hughes et al., 2005], network flow problems [Balakrishnan and Graves, 1989], and other applications.

For such problems, we would like to compute gradients "through" operations involving spline approximation. However, algorithms for spline approximation often involve discontinuous (or even discrete) co-domains and may introduce undefined (or even zero) gradients. Generally, embedding such functions as layers in a differentiable program, and running automatic differentiation on this program, requires special care. A popular solution is to relax these non-differentiable, discrete components into continuous approximations for which gradients exist. This has led to recent advances in differentiable sorting [Blondel et al., 2020; Cuturi et al., 2019], dynamic programming [Mensch and Blondel, 2018], and optimization [Djolonga and Krause, 2017; Agrawal et al., 2019; Deng et al., 2020].

**Our contributions:** We propose a principled approach for differentiable programming for spline functions *without* the use of continuous relaxation[2]. For the forward pass, we leverage fast algorithms for computing the optimal projection of any given input onto the space of piecewise polynomial functions. For the backward pass, we leverage a fundamental *locality* property in splines that every piece (or basis function) in the output approximation only interacts with a few other elements. Using this, we derive a weak form of the Jacobian for the spline operation and show that it exhibits a particular block-structured form. While we focus on spline approximation in this paper, our approach can be generalized to any algorithmic module with piecewise smooth outputs. Our specific contributions are as follows:

1. We propose the use of spline function approximations as "layers" in differentiable programs.
2. We derive efficient (nearly-linear time) methods for computing forward and backward passes for various spline approximation problems, showing that the (weak) Jacobian in each case can be represented using a *block sparse* matrix that can be efficiently used for backpropagation.
3. We show applications of our approach in three stylized applications: image segmentation, 3D point cloud reconstruction, and finite element analysis for the solution of partial differential equations.

**Related Work** Before proceeding, we briefly review related work.

**Extensions of autodiff:** Automatic differentiation (autodiff) algorithms enable gradient computations over basic algorithmic primitives such as loops, recursion, and branch conditions [Baydin et al., 2018]. However, introducing more complex non-differentiable components requires careful treatment due to undefined or badly behaved gradients. For example, in the case of sorting and ranking operators, it can be shown that the corresponding gradients are either uninformative or downright pathological, and it is imperative the operators obey a 'soft' differentiable form. Cuturi et al. [2019] propose a differentiable *proxy* for sorting based on optimal transport. Blondel et al. [2020] improve this by proposing a more efficient differentiable sorting/ranking operator by appealing to isotonic regression. Berthet et al. [2020] introduce the use of stochastic perturbations to construct smooth approximations to discrete functions, and other researchers have used similar approaches to implement end-to-end trainable top-$k$ ranking systems [Xie et al., 2020; Lee et al., 2020]. Several approaches for enabling autodiff in optimization have also been researched [Pogančić et al., 2020; Amos and Kolter, 2019; Agrawal et al., 2019; Mensch and Blondel, 2018].

**Structured priors as neural "layers":** As mentioned above, one motivation for our approach arises from the need for enforcing structural priors for scientific computing applications. Encoding non-differentiable priors such as the solutions to specific partial differential equations [Sheriffdeen et al., 2019], geometrical constraints [Joshi et al., 2020; Chen et al., 2019], and spatial consistency measures [Djolonga and Krause, 2017] perform well but typically require massive amounts of structured training examples.

**Spline approximation:** Non-Uniform Rational B-splines (NURBS) are commonly used for defining spline surfaces for geometric modeling [Piegl and Tiller, 1997]. NURBS surfaces offer a high level of control and versatility; they can also compactly represent the surface geometry. The versatility of NURBS surfaces enables them to represent more complex shapes than Bèzier or B-splines. Several frameworks that leverage deep learning are beginning to use NURBS representations. Minto et al. [2018] use NURBS surfaces fitted over the 3D geometry as an input representation for the object

---

[2]While tricks such as straight-through gradient estimation [Bengio, 2013] also avoid continuous relaxation, they are heuristic in nature and may be inaccurate for specific problem instances [Yin et al., 2019].

classification task of ModelNet10 and ModelNet40 datasets. Erwinski et al. [2016] presented a neural-network-based contour error prediction method for NURBS paths. Fey et al. [2018] present a new convolution operator based on B-splines for irregular structured and geometric input, e.g., graphs or meshes. Very recently, Sharma et al. [2020] perform point cloud reconstruction to predict a B-spline surface, which is later processed to obtain a complete CAD model with other primitives "stitched" together.

**Differentiable PDE solvers:** With the advent of deep learning, there has been a recent rise in the development of differentiable programming libraries for physics simulations [Hu et al., 2019; Qiao et al., 2020]. Most often, the physics phenomena are represented using partial differential equations (PDEs) [Sanchez-Gonzalez et al., 2020; Holl et al., 2020]. Considerable effort has gone into designing physics-informed loss functions [Raissi et al., 2019; Raissi and Karniadakis, 2018; Kharazmi et al., 2021] whose optimization leads to desired solutions for PDEs. Due to space limitations, we defer to a detailed survey of this (vast) area by Cai et al. [2021].

## 2 Differentiable Spline Approximation

We now introduce our framework, Differentiable Spline Approximation (DSA), as an approach to estimate gradients over piecewise polynomial operations. Our main goal will be to estimate easy-to-compute forms of the (weak) Jacobian for several spline approximation problems, enabling their use within backward passes in general differentiable programs.

**Setup.** We begin with some basic definitions and notation. Let $f \in \mathbb{R}^n$ be a vector where the $i^{\text{th}}$ element is denoted as $f_i$. Let us use $[n] = \{1, 2, \ldots, n\}$ to denote the set of all coordinate indices. For a vector $f \in \mathbb{R}^n$ and an index set $I \subseteq [n]$, let $f_I$ be the restriction of $f$ to $I$, i.e., for $i \in I$, we have $f_I(i) := f_i$, and $f_I(i) := 0$ for $i \notin I$. Now, consider any fixed partition of $[n]$ into a set of disjoint intervals $\mathcal{I} = \{I_1, \ldots, I_k\}$ where the number of intervals $|\mathcal{I}| = k$. The $\ell_2$-norm of $f$ is written as $\|f\|_2 := \sqrt{\sum_{i=1}^n f_i^2}$ while the $\ell_2$ distance between $f, g$ is written as $\|f - g\|_2$.

We first define the notion of a *discretized $k$-spline*. Note that the use of "spline" here is non-standard and somewhat more general than what is typically encountered in the literature. (Indeed, the spline concept used in computer graphics is a special instance of this definition; we explain further below.)

**Def. 2.1** (Discretized $k$-spline). A vector $h \in \mathbb{R}^n$ is called a discretized $k$-spline with degree $d$ if: (i) there exists a partition of $[n]$ into $k$ disjoint intervals $I_1, \ldots, I_k$; (ii) within each interval $I_i$, the coefficients of $h_j$, $j \in I_i$, can be perfectly interpolated by some polynomial function of degree $d$.

Let us illustrate this by an example. Suppose that $d = 1$ and $k = 5$. Then, $h$ is a discretized $k$-spline with degree $d$ if, in a "line plot" of the vector $h$ (i.e., we interpolate the 2D points $(j, h_j)$ for all $j \in [n]$), we see up to $k = 5$ distinct linear pieces. A different way to interpret this definition is that we start with a piecewise degree-$d$ polynomial function $H : \mathbb{R} \to \mathbb{R}$ with $k = 5$ pieces (with suitably defined knot points, which are the location of the intervals $I$), and evaluate $H$ at any $n$ equally spaced points in its domain. This gives us a vector $h \in \mathbb{R}^n$, which we call a discretized $k$-spline. In contrast with traditional splines, we allow $H$ to be arbitrarily defined *at* the knot points and require no specific continuity or differentiability properties. Therefore, our definition encompasses all standard spline families (including interpolating/approximating splines such as smoothing-, cubic-, and B-splines).

### 2.1 Spline Approximation

Our focus in this paper is the problem of computing the best possible spline fit to a given set of data points (where both the parameters of the spline as well as the knot vectors are allowed to be variable).

We provide an algebraic interpretation of this problem. For a given vector space $\mathbb{R}^n$, consider $S_d^k$, the set of all discretized $k$-splines with degree $d$. Since (standard) splines are vector spaces for a fixed set of knots, one can easily see that for any fixed partition of $[n]$ into $k$ subsets, the family of discretized $k$-splines is a $k(d + 1)$-dimensional subspace of $\mathbb{R}^n$. Now suppose that the knot indices are allowed to vary. The number of possible partitions is finite (of the order of $\binom{n}{k}$), and therefore the set $S_d^k$ is a *finite union of subspaces*, or a nonlinear submanifold, embedded in $\mathbb{R}^n$.

Therefore, the problem of discretized $k$-spline approximation can be viewed as an *orthogonal projection* onto this nonlinear manifold. Consider any arbitrary vector $x \in \mathbb{R}^n$ (we can think of $(i, x_i)$

as a set of $n$ data points to which we are trying to fit a $k$-spline). Then, the best $k$-spline fit to $x$ (in the sense of $\ell_2$ distance) amounts to solving the optimization problem:

$$F(x) = \arg\min_h \frac{1}{2}\|x - h\|_2^2 = \frac{1}{2}\sum_{i=1}^n (x_i - h_i)^2 \text{ s.t. } h \in S_d^k \tag{1}$$

This operation resembles standard spline regression. But it is strictly more general since this requires not only optimizing piecewise spline parameters but *also the knot indices*. Crucially, we note that $F$ is both a non-differentiable and a non-convex map. Nevertheless, such an orthogonal projection can be computed in polynomial (in fact, nearly-linear) time [Jagadish et al., 1998; Acharya et al., 2015] using many different techniques, including dynamic programming. This forms the *forward pass* of our DSA "layer".

Our first main conceptual contribution is a formal derivation of the *backward pass* of the orthogonal projection operation. Strictly speaking, the Jacobian is not well-defined due to the non-differentiable nature of the forward pass (owing to the non-differentiability built into the definition of the $k$-spline). Therefore, we will instead be deriving the so-called "weak" form of the Jacobian (borrowing terminology from Blondel et al. [2020]).

We leverage two properties of the projection operation: (1) the output of the forward pass $h$ corresponds to a partition of $[n]$, that is, each element of $h_j$ corresponds to a *single* interval, $I_j$, and (2) within each interval, the least-squares operation is continuous and differentiable. The first property ensures that every element $x_i$ contributes to only a single piece in the output $h$. Given that the sub-functions from the piecewise partitioning function are smooth, we also observe that the size of each block corresponds to the size of the partition, $I_i$. Using this observation, we get:

**Theorem 1.** The Jacobian of the operation $F$ with respect to $x \in \mathbb{R}^n$ can be expressed as a *block diagonal* matrix, $\mathbf{J} \in \mathbb{R}^{n \times n}$, whose $(s, t)^{\text{th}}$ entry obeys:

$$\mathbf{J}_x(F(x))(s, t) = \frac{\partial h(x)_s}{\partial x_t} = \begin{cases} \frac{\partial h_{I_i}(x)_s}{\partial x_t} & \text{if } s, t \in I_i \\ 0 & \text{otherwise} \end{cases} \tag{2}$$

As a concrete instantiation of this result, consider the case $d = 0$. This is the case where we wish to best approximate the entries of $x$ with at most $k$ "horizontal" pieces, where the break-points are obtained during the forward pass[3]. Call this approximation $h$. Then, the Jacobian of $h$ with respect to $x$ forms the block-diagonal matrix $\mathbf{J} \in \mathbb{R}^{n \times n}$:

$$\mathbf{J} = \begin{bmatrix} \mathbf{J}_1 & \mathbf{0} & \dots & \mathbf{0} \\ \mathbf{0} & \mathbf{J}_2 & \dots & \mathbf{0} \\ \vdots & \vdots & \ddots & \vdots \\ \mathbf{0} & \mathbf{0} & \dots & \mathbf{J}_k \end{bmatrix} \tag{3}$$

where all entries of each block, $\mathbf{J}_i \in \mathbb{R}^{|I_i| \times |I_i|}$ are *constant* and equal to $1/|I_i|$, i.e., they are row/column-stochastic. Note that the sparse structure of the Jacobian allows for fast computation and that computing the Jacobian vector product $\mathbf{J}^T \nu$ for any input $\nu$ requires $O(n)$ running time. As an additional benefit, the decoupling induced by the partition enables further speed up in computation via parallelization. See the Appendix for proofs, as well as derivations of similar Jacobians for $k$-spline approximation of any degree $d \geq 1$, and generalization to 2D domains (surface approximation). In Section 3 we demonstrate the utility of this approach for a 2D segmentation (i.e., piecewise constant approximation) problem, similar to the setting studied in Djolonga and Krause [2017].

---

[3]In the data summarization literature, this class of functions is sometimes called $k$-histograms [Jagadish et al., 1998]

## 2.2 Differentiable NURBS

We now switch to a slightly different setting involving a special spline family known as non-uniform rational B-splines (NURBS), which are common in geometric modeling. Mathematically, a NURBS curve is a continuous function $\mathbf{C} : \mathbb{R} \to \mathbb{R}$ defined as follows. Construct any knot vector $u$ (i.e. a non-decreasing sequence of real coordinate values) and fix degree $d$. Recursively define a sequence of basis functions, $N_i^d : \mathbb{R} \to \mathbb{R}$ computed using the *Cox-de Boor formula*:

$$N_i^d(u) = \frac{u - u_i}{u_{i+d} - u_i} N_i^{d-1}(u) + \frac{u_{i+d+1} - u}{u_{i+d+1} - u_{i+1}} N_{i+1}^{d-1}(u), \ N_i^0(u) = \left\{ \begin{array}{ll} 1 & \text{if } u_i \leq u \leq u_{i+1} \\ 0 & \text{otherwise} \end{array} \right. \tag{4}$$

for $d = 1, 2, \ldots$. In the uniform case (where the knots are equally spaced), each $N_i^d$ can be viewed as being generated by recursively convolving a box function with $N_i^{d-1}$. The non-uniform case cannot be written as a convolution, but the intuition is similar. With these basis functions in hand, the NURBS curve $\mathbf{C}$ is defined as the rational function:

$$\mathbf{C}(u) = \frac{\sum_{i=0}^{n} N_i^d(u) w_i \mathbf{P}_i}{\sum_{i=0}^{n} N_i^d(u) w_i}, \tag{5}$$

where $P_i$, $i = 0, 1, \ldots, t$ are called *control points* and $w_i$ are corresponding non-negative weights. The number of control points is related to the number of knots $k$ and curve degree $d$ as follows: $k = t + d + 1$. For simplicity, assume that all weights are equal to one. The basis functions in NURBS add up to one uniformly for each $u$ (this is called the *partition of unity* property). Therefore:

$$\mathbf{C}(u) = \sum_{i=0}^{t} N_i^d(u) \mathbf{P}_i, \tag{6}$$

In summary, the NURBS curve is parametrically defined via the control points and the knot positions. This discussion is for 1D curves, but an extension to higher-order surfaces is conceptually similar.

Consider implementing NURBS as a differentiable "layer" where the inputs are the knot positions and control points. The forward pass through this layer simply consists of evaluating Equation 6 via the recursive Equation 4, and storing the various basis functions (and their spans) for further use.

However, the backward pass is a bit more complicated, once again due to the *non-differentiable* nature of $\mathbf{C}$. The gradient with respect to the control point coordinates, $\mathbf{P}$ is straightforward since the mapping from $\mathbf{P}$ to $\mathbf{C}$ is linear. However, the gradient with respect to the *knot* positions, $u_i$, is not well-defined due to the non-differentiable nature of the *base cases* of the recursion (which are box functions specified in terms of $u_i$). Once again, we see that the non-differentiability of NURBS is built into its very definition, and this affects the numerics.

To resolve this, we propose the following approach to compute an (approximate) Jacobian of $\mathbf{C}$. The main source of the issue is the derivative of the box-car function $N_i^0(u) = \mathbf{1}_{[u_i, u_{i+1}]}$ with respect to the knot points, which is not well defined. However, $N_i^0(u)$ can be viewed as the difference between convolutions of the unit step function with $\delta_{u_i}$ and $\delta_{u_{i+1}}$, where $\delta$ is the Dirac delta defined over the real line. We smoothly approximate the delta function by a Gaussian function with small enough bandwidth hyperparameter $\sigma$: $\delta(u_i) \approx g(u) = \exp(-(u - u_i)/2\sigma^2)$. This function is now differentiable with respect to $u_i$, with $g'(u) = \frac{u - u_i}{\sigma^2} g(u)$. Convolutions and differences are linear, and hence the derivative is the basis function times a multiplicative factor. Finally, a similar approach as the Cox-de Boor recursion (Equation 4) can be used to reconstruct the derivatives for all basis functions of higher order. See Algorithm 1 for pseudocode and the Appendix for details.

---

**Algorithm 1** Backward pass for NURBS Jacobian (for one curve point , $\mathbf{C}(u)$)

---

$\mathbf{P}'$, $\mathbf{U}'$: gradients of $\mathbf{C}$ w.r.t. $\mathbf{P}$, $\mathbf{U}$
Initialize: $\mathbf{P}', \mathbf{U}' \to 0$
Retrieve $u_{span}, N_i^d, \mathbf{C}(u)$ calculated during forward pass
/* $u_{span}$ is the index of knot position */
/* $N_i^d$ is the basis function of degree $d$ */
/* $\mathbf{C}(u)$ is the evaluated curve point */
**for** $h = 0 : d + 1$ **do**
    $\mathbf{P}'_{u_{span}+h} = N_h^d$ // easy since $\mathbf{C}$ is a linear function of $\mathbf{P}$.
    $\mathbf{U}'_{u_{span}+h} = N_h^d \mathbf{U}_{u_{span}+h}$ // due to Gaussian approximation; see discussion below.

---

Let us probe the structure of this Jacobian a bit further. Suppose we evaluate the curve $\mathbf{C}$ at $n$ arbitrary domain points. There are slightly less than $k$ control points, and therefore the Jacobian is roughly of size $n \times O(k)$. However, due to the recursive nature of the definition of basis functions, the *span* (or support) of each basis function is small and only touches $d + 1$ knots; for example, only 2 knots affect $N_i^0$, only 3 knots impact $N_i^1$, and so on. This endows a natural sparse structure on the Jacobian. Moreover, for a fixed order parameter $d + 1$, the span is constant [Piegl and Tiller, 1997]; therefore, assuming evenly spaced evaluation points, we have the same number of nonzeros. Therefore, the Jacobian exhibits an interesting *Toeplitz* structure (unlike the block diagonal matrix in the case of Equation 3), thereby enabling efficient evaluation during any gradient calculations. We show below in Section 3 that automatic differentiation using this approach surpasses existing NURBS baselines.

## 2.3 Differentiable Finite Element PDE Solvers

Next, we see how spline approximations can be used to improve finite element analysis for solving PDEs. Popular recent efforts for solving PDEs using autodiff construct "physics-informed" solvers [Raissi et al., 2019; Raissi and Karniadakis, 2018], while other efforts have been made to utilize variational [Kharazmi et al., 2021] or adjoint-based derivative methods [Holl et al., 2020]. However, these approaches come with challenges while used in conjunction with autodiff packages, and gradient pathologies pose a major barrier [Wang et al., 2020].

Using our principles developed above, we propose an alternative PDE solution approach via *differentiable finite elements*. PDE solvers based on Finite Element Methods (FEM) are ubiquitous, and we provide a very brief primer here. Consider a domain $\Omega$ and a differential system of equations:

$$\mathcal{N}[\mathbf{U}(\underline{u})] = F(\underline{u}), \quad \underline{u} \in \Omega, \tag{7}$$

where $\mathcal{N}$ denotes the differential operator and $\mathbf{U} : \Omega \rightarrow \mathbb{R}$ is a continuous field variable; it is common to specify additional boundary constraints on $\mathbf{U}$. The *Galerkin method* converts solving for the best possible $\mathbf{U}$ (which is a continuous variable) into a discrete problem by first looking at the weak form: $R(\mathbf{U}) = \int_\Omega V [\mathcal{N}(\mathbf{U}) - F] \, d\underline{u}$, where $V$ is called a *test function* (and the weak form may involve some integration by parts), and rewriting this weak form in terms of a finite set of basis coefficients. A typical set of basis functions $\Phi_j$ is obtained by (piecewise) concatenation of polynomials, each defined over elements of a given partition of $\Omega$ (also called a *mesh*). Commonly used choices include Lagrange polynomials, defined by:

$$p_{i,d}^r(\underline{u}) = \sum_{r=1}^{d} \mathbf{U}_r \prod_{\substack{0 \leq m \leq d \\ m \neq r}} \frac{\underline{u} - u_m}{u_r - u_m} \text{ s.t. } x_r \in [-1, 1] \tag{8}$$

where $\{u_0, u_1, \ldots, u_d\}$ are a finite set of nodes (akin to control points in our above discussion, except in this case the splines interpolate the control points) and $\mathbf{U}_r$ is the corresponding coefficient. We use this collection of basis functions $\Phi_j$ to represent $\mathbf{U}$:

$$\mathbf{U}(\underline{u}) = \sum_{j=1}^{\#\text{nodes}} \Phi_j(\underline{u}) \mathbf{U}_j^d \tag{9}$$

and likewise for $V$. (The resemblance with Equation 5 above should be clear, and indeed NURBS basis functions could be an alternative choice.) Plugging the discrete coefficient representation $\mathbf{U}^c := \{\mathbf{U}_j^c\}$ into the definition of $R$, we get a standard Finite Element form,

$$R(\mathbf{U}^c, V^c) = B(\mathbf{U}^c, V^c) - L(V^c) \tag{10}$$

where $B(\mathbf{U}^c, V^c)$ is the discrete form (bilinear for linear operators) that encodes the differential operator and $L(v)$ is a linear functional involving the forcing function. For most PDE operators (including linear elliptic operators), one can form the *energy functional* by using $U$ as the test function:

$$J(\mathbf{U}^c) = \frac{1}{2} B(\mathbf{U}^c, \mathbf{U}^c) - L(\mathbf{U}^c). \tag{11}$$

Optimization of this energy functional can now be performed using gradient-based iterations evaluated by automatic differentiation. This is a powerful approach since formal techniques exist (e.g., Galerkin Least Squares Bochev and Gunzburger [2009]) that reformulate weak forms of PDEs into equivalent energy functionals. The key aspect to note here is that differentiating "through" the differential operator $\mathcal{N}$ (embedded within $B$) requires derivative computations of the piecewise polynomial basis functions $\Phi_j$s, and therefore our techniques developed above are applicable.

# 3 Experiments

We have implemented the DSA framework (and its different applications provided below) by extending `autograd` functions in Pytorch. We also provide the capability to run the code using CUDA for GPU support. All the experiments were performed using a local cluster with 6 compute nodes and each node having 2 GPUs (Tesla V100s with 32GB GPU memory). All training was done using a single GPU. We summarize all our experiments in Table 1. Each experiment shown below is performed multiple times with different random seeds, and the average value with error bars is provided. Due to limited space, we provide three interesting applications of spline approximations here (see Appendix for additional examples). We have also open-sourced the code at https://github.com/idealab-isu/DSA.

**Image segmentation:** We begin with implementing a 2D piecewise constant splines regression approach for the image segmentation problem using a UNet [Ronneberger et al., 2015]. For differentiation, we use the formulation of splines discussed in Section 2. We analyze the efficacy of our approach by adding a piecewise constant DSA layer as the final layer of our network ($M_{\text{DSA}}$). We compare this approach with the baseline model without the piecewise constant layer ($M_{\text{baseline}}$).

We train two models ($M_{\text{DSA}}$, $M_{\text{baseline}}$) on two different segmentation tasks: the Weizmann horse dataset [Borenstein and Ullman, 2004] and the Broad Bioimage Benchmark Collection dataset [Ljosa et al., 2012] (publicly available under Creative Commons License). We split both the Weizmann horse and Broad Bioimage Benchmark Collection datasets into train and test with 85% and 15% of the dataset. We use binary cross-entropy error between the ground truth and the predicted segmentation map. We use the same architecture and hyper-parameters for both models (see Appendix for details.)

We observe that our DSA layer provides more consistent segmentation maps and higher Jaccard scores than the baseline model; see Figure 1. For the Weizmann horse dataset, $M_{\text{conn}}$ enforces the connectivity of the segmented objects while also limiting noise in the segmentation map. In the cell

Table 1: ***Summary of Experiments:*** *We present three experiments in this paper with diverse applications, model architectures, basis functions, and degree of splines.*

| Application | Input | Output | Architecture | Spline Function | Degree |
|---|---|---|---|---|---|
| Image Segmentation | Image | 2D piecewise constant knot partitions | U-Net | Box-Car functions | 0 |
| Point cloud reconstruction | 3D point cloud | 3D control points and rational weights | Dynamic Graph CNN | BSpline polynomials | 3 |
| PDE-based surrogate physics | 2D Mesh (regular grid) | 2D physics field on the mesh grid | U-Net | Lagrange polynomials | 1,2,3 |

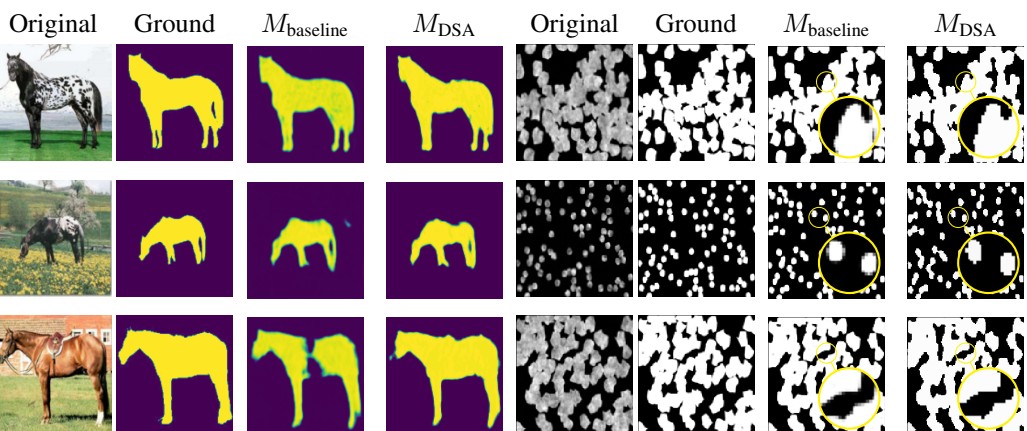

Figure 1: ***Segmentation results.*** *The two models, $M_{DSA}$ and $M_{baseline}$ were trained with and without the DSA layer, respectively. Note that $M_{DSA}$ generates better segmentation masks with fewer holes and enforced connectivity. Note the sharper edges compared to the standard segmentation results. Additional figures are in the Appendix.*

Table 2: ***Results for the horse and cell segmentation dataset:*** *Jaccard scores for the baseline and connected component models for the cell and horse segmentation task. From independent three runs with random seeds and the table reports mean and standard deviation. As the objects of interest (piecewise constant components) are smaller, the model with the DSA layer learns a better representation. Predictions are thresholded at* 0.5.

| Dataset | Baseline ($M_{\text{baseline}}$) | Baseline + DSA ($M_{\text{DSA}}$) |
|---|---|---|
| Weizmann Horse [Borenstein and Ullman] | $72.06 \pm 0.60$ | $\mathbf{73.13 \pm 0.31}$ |
| Broad Bioimage Benchmark [Ljosa et al.] | $79.34 \pm 0.43$ | $\mathbf{81.56 \pm 0.24}$ |

segmentation task, we note that the number of segments is high while the objects are small. Since the size of the components is small, our DSA layer Jacobian exhibits substantial differences from the commensurate identity gradient for the baseline models. Table 2 also shows the further improvement in Jaccard score on cell segmentation tasks over the Weizmann horse dataset.

**3D point cloud reconstruction using NURBS:** Next, we provide results for two experiments using DSA with NURBS discussed in Section 2.2. The first application is surface fitting for a complex benchmark surface represented by a mesh of surface points obtained by evaluating the benchmark test function at these points. We use Bukin function N.6 (publicly available here) for generating a grid of $256 \times 256$ points as shown on the left of Figure 2. For fitting a NURBS surface from the defined target point cloud, we initialize a uniform clamped knot vector for a cubic basis function and random control points of size $8 \times 8$. Using DSA, we evaluate the NURBS surface for a uniform grid of $256 \times 256$ parametric points. We now evaluate the surface and use mean squared error for fitting the surface point cloud using NURBS. We consider two scenarios: (i) we do not update the knot vectors (i.e., no reparameterization), and (ii) we compute the gradients for the knot vectors and allow for reparameterization (i.e., change of knot locations). We provide the comparison of these scenarios in Table 3. We see that the reparameterization helps in reducing the error in fit by half. Also, we notice that the density of points evaluated has a very minimal impact on the performance (see more details in Appendix). Visually, in Figure 2, we see that two knots in the "$v$" direction come close to each other around 0.06, enabling a sharp edge in the evaluated surface.

The next experiment we present involves surface reconstruction from point clouds using a graph convolutional neural network and DSA for unsupervised training. We use the SplineNet method proposed by Sharma et al. [2020] to be the baseline for point cloud reconstruction using splines. SplineNet uses a dynamic graph convolutional neural network (DGCNN) to predict the control points for a spline surface. The authors use a supervised control point loss to perform the training and include regularizations such as the Laplacian loss and a patch distance (using Chamfer distance) loss. Instead, we perform this training in an unsupervised manner by not using the control points prediction loss and only using DSA to evaluate the surface and then apply regularization of minimizing the Laplacian of the surface. Since we can train this unsupervised, we can even use an arbitrary number of control points and are not restricted to the target control points.

For a fair comparison, we use the same network, dataset, and hyperparameters as Sharma et al. [2020] and change the loss functions by removing the control point regression loss. For comparison, we compute the chamfer distance between the input point cloud and the NURBS surface fit by the DGCNN model ($M_{DSA}$) (see Appendix for details of training). We use the Spline Dataset, which is

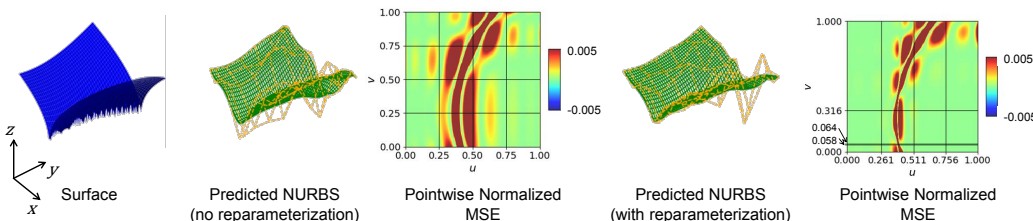

Figure 2: ***NURBS surface fitting results:*** *Surface fitting to point cloud generated using the Bukin's function N.6 given by $z = 100\sqrt{|y - 0.01x^2|} + 0.01|x + 10|; -15 < x < -5, -3 < y < 3$. The center image shows the surface fit obtained without reparameterization of the knots. We obtain better fit by reparameterizing the knots.*

Table 3: **NURBS surface fitting results:** *Comparison of mean squared error between the target surface point cloud and the surface generated using DSA with and without reparameterization.*

| Number of Points | $M_{DSA}$(without reparameterization) | $M_{DSA}$(with reparameterization) |
|---|---|---|
| $128 \times 128$ | $19.83 \pm 0.001$ | $\mathbf{8.25 \pm 0.01}$ |
| $256 \times 256$ | $19.85 \pm 0.001$ | $\mathbf{8.23 \pm 0.02}$ |

Table 4: **Point-cloud reconstruction results:** *Comparison between the model proposed by Sharma et al. [2020] and its extension using DSA (with different number of control points). We compare the two-sided chamfer distance (scaled by 100) between the input point cloud and the fitted surface.*

| Experiment | $M_{baseline}$ $(20 \times 20)$ | $M_{DSA}$ $(20 \times 20)$ | $M_{DSA}$ $(5 \times 5)$ | $M_{DSA}$ $(4 \times 4)$ |
|---|---|---|---|---|
| **Chamfer Distance** | $1.18 \pm 0.10$ | $0.03 \pm 0.02$ | $0.14 \pm 0.07$ | $\mathbf{0.02 \pm 0.01}$ |

a subset of surfaces extracted from the ABC dataset (available for public use under this license). In Table 4, we provide a comparison of chamfer distance obtained between the predicted surface points from splines and the input point cloud for the test dataset. In our experiments, we observe that we get significantly better performance with fewer control points. This is because most of the surfaces in the dataset are simple curved surfaces that can be easily fit with fewer control points.

**PDE based surrogate physics priors:** Finally, we leverage DSA in the context of solving PDEs as a prior. In particular, we consider the Poisson equation solved for $u$:

$$-\nabla \cdot (\nu(\mathbf{x})\nabla u) = f(\mathbf{x}) \text{ in } D \tag{12}$$
$$u|_{\partial D} = 0 \tag{13}$$

where $D = [0, 1]^2$, a 2D square domain, $\nu$ is the *diffusivity* and $f$ is the forcing function. We consider two experiments here: (1) validation of our approach with an analytically known solution, and (2) extending this to learn the solutions for the parametric Poisson equation parameterized using $\nu$.

For the first experiment, we set $\nu$ to 1 and the forcing $f = f(\underline{x}) = f(x, y) = 2\pi^2 \sin(\pi x)\sin(\pi y)$, and minimize the residual using the approach described in Section 2.3. We know that for this PDE and the conditions provided, the exact solution is given by $u_{ex}(x, y) = \sin(\pi x)\sin(\pi y)$. We compare our results ($u_{DSA}$) with the exact solution $u_{ex}$. Also, we perform this experiment with Lagrange polynomials of different degrees. Further, we compare our results with results obtained using PINNs [Raissi et al., 2019]. We obtain significantly better performance (lesser $\ell_2$-error by an order of magnitude) compared to PINNs, owing to more accurate gradients computed using our DSA approach. The performance improvement with increase in degree of polynomial in lower resolutions is more pronounced than at higher resolutions.

Next, we present results for training a deep learning network with a prior for solving a *parametric* Poisson's equation. The input to the network are different diffusivity maps $\nu$ sampled from

$$\nu(\mathbf{x}; \omega) = \exp \left( \sum_{i=1}^{m} \omega_i \lambda_i \xi_i(x)\eta_i(y) \right) \tag{14}$$

where $\omega_i$ is an $m$-dimensional parameter, $\lambda$ is a vector of real numbers with monotonically decreasing values arranged in order; and $\xi$ and $\eta$ are functions of $x$ and $y$ respectively. We take $m = 4$, $\omega = [-3, 3]^4$ and $\lambda_i = \frac{1}{(1+0.25a_i^2)}$, where $\mathbf{a} = (1.72, 4.05, 6.85, 9.82)$. Also $\xi_i(x) = \frac{a_i}{2}\cos(a_i x) + \sin(a_i x)$ and $\eta(y) = \frac{a_i}{2}\cos(a_i y) + \sin(a_i y)$. We generate several diffusivity maps by sampling this function with different values of $\omega$.

Table 5: **Quantitative comparison of Solving PDEs:** $L_2$ *Norm between the analytical exact solution* $u_{ex}$ *and predicted* $u$ *using PINNs [Raissi et al., 2019] and DSA with different degrees of the Lagrange polynomials.*

| Model | | $PINN$ | $DSA\,(d=1)$ | $DSA\,(d=2)$ | $DSA\,(d=3)$ |
|---|---|---|---|---|---|
| $L_2$ Norm | $128 \times 128$ | $3.72 \pm 0.20$ E-4 | $3.32 \pm 0.05$ E-5 | $\mathbf{2.16 \pm 0.04}$ **E-5** | $2.37 \pm 0.10$ E-5 |
| | $256 \times 256$ | $2.63 \pm 0.20$ E-4 | $\mathbf{2.57 \pm 0.01}$ **E-5** | $2.79 \pm 0.20$ E-5 | $2.59 \pm 0.10$ E-5 |

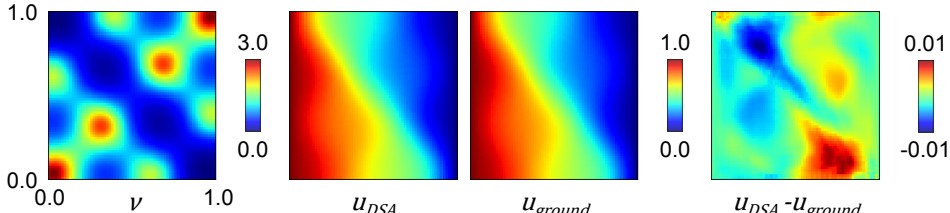

Figure 3: ***Learning a parametric family of PDE solutions:*** *Poisson's equation with log permeability coefficients* $\omega = (-0.26, -0.77, -0.37, -0.92)$ *in the Poisson's equation.*

We use a UNet [Ronneberger et al., 2015] that takes these diffusivity maps and predicts the solution $u$, which is further optimized with the residual minimizing prior to the Poisson's equation. Thus, we obtain a trained neural network that predicts the solution field $u$ for any unknown diffusivity maps from the data distribution. We provide the predicted result along with its comparison with traditional numerical FEM results in Figure 3. Visually, we see both the predicted solution field map ($u_{DSA}$) and the actual solution field ($u_{ground}$) obtained using traditional numerical methods match each other. The right most image shows the difference between both with the maximum deviation to be $0.01$, showing the accuracy of our (easy-to-implement) DSA-based FEM solver.

## 4 Broader Impact and Discussion

We introduce a principled approach to estimate gradients for spline approximations. Specifically, we derive the (weak) Jacobian in the form of a block-sparse matrix based on the partitions generated by any spline approximation algorithm (which serves as the forward pass). The block structure allows for fast computation of the backward pass, thereby extending the application of differentiable programs (such as deep neural networks) to tasks involving splines. Our methods show superior performance than the state-of-the-art curve fitting methods by reducing the chamfer distance by an order of magnitude and the mean squared error in the case of surface fitting by a factor of two. Further, with the application of our methods in finite element analysis, we show significantly better performance than state-of-the-art physics-informed neural networks.

Our method is quite generic and may impact applications such as computer graphics, physics simulations, and engineering design. Care should be taken to ensure that these applications are deployed responsibly. Future works include further algorithmic understanding of the inductive bias encoded by DSA layers and dealing with splines having a dynamically chosen number of parameters (control points and knots).

## Acknowledgements

This work was supported in part by the National Science Foundation under grants CCF-2005804, LEAP-HI:2053760, CMMI:1644441, CPS-FRONTIER:1954556, USDA-NIFA:2021-67021-35329 and ARPA-E DIFFERENTIATE:DE-AR0001215. Any information provided and opinions expressed in this material are those of the author(s) and do not necessarily reflect the views of, nor any endorsements by, the funding agencies.

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
