# Appendix

## A Proofs and derivations

**Theorem 1.** The Jacobian of the operation $F$ with respect to $x \in \mathbb{R}^n$ can be expressed as a *block diagonal* matrix, $\mathbf{J} \in \mathbb{R}^{n \times n}$, whose $(s,t)^{\text{th}}$ entry obeys:

$$\mathbf{J}_x(F(x))(s,t) = \frac{\partial h(x)_s}{\partial x_t} = \begin{cases} \frac{\partial h_{I_i}(x)_s}{\partial x_t} & \text{if } s,t \in I_i \\ 0 & \text{otherwise} \end{cases} \tag{15}$$

*Proof.* The proof follows similar arguments as in Proposition 4 from Blondel et al. [2020].

Let $\mathcal{I} = \{I_1, I_2, \cdots, I_k\}$ be $k$ partitions induced by some $H : \mathbb{R} \to \mathbb{R}$ for some input, $\mathbf{x} \in \mathbb{R}^n$ and $h \in \mathbb{R}^n$ be a vector from $n$ equally spaced evaluated $H$ in its domain. Then, each element, $x_i$ uniquely belongs to some partition $I_r$.

Now,

$$\mathbf{J}_x(F(x))(s,t) = \frac{\partial \sum_{j=1}^{k} h(x)_s \odot \mathbb{1}(s \in I_j)}{\partial x_t}$$

$$= \begin{cases} \frac{\partial h(x)_s}{\partial x_t} & \text{if } s,t \in I_r \\ 0 & \text{otherwise} \end{cases}$$

Note that this is a block-diagonal matrix with each block being $|I_r| \times |I_r|$, giving us the required statement. $\square$

## B Application of DSA to piecewise polynomial regression

**1D piecewise constant regression:** We first provide the notations we provided in Section 2.

Let $f \in \mathbb{R}^n$ be a vector where the $i^{\text{th}}$ element is denoted as $f_i$. Let us use $[n] = \{1, 2, \ldots, n\}$ to denote the set of all coordinate indices. For a vector $f \in \mathbb{R}^n$ and an index set $I \subseteq [n]$, let $f_I$ be the restriction of $f$ to $I$, i.e., for $i \in I$, we have $f_I(i) := f_i$, and $f_I(i) := 0$ for $i \notin I$. Now, consider any fixed partition of $[n]$ into a set of disjoint intervals $\mathcal{I} = \{I_1, \ldots, I_k\}$ where the number of intervals $|\mathcal{I}| = k$. The $\ell_2$-norm of $f$ is written as $\|f\|_2 := \sqrt{\sum_{i=1}^{n} f_i^2}$ while the $\ell_2$ distance between $f, g$ is written as $\|f - g\|_2$. Finally, $\mathbb{1}_I \in \{0, 1\}^n$ is a indicator vector where for $i \in I$, $\mathbb{1}_I(i) = 1$ and for $i \notin I$, $\mathbb{1}_I(i) = 0$.

We consider the case of $k$-piecewise regression in 1D, where we can use any algorithm to approximate a given input vector with a fixed number of piecewise polynomial functions. The simplest example is that of $k$-piecewise *constant* regression, where a given input vector is approximated by a set of constant segments.

Formally, consider a piecewise constant function $H : \mathbb{R} \to \mathbb{R}$ with $k$ pieces. Similar to spline, we evaluate $H$ at any $n$ equally spaced points in its domain. This gives us a vector $h \in \mathbb{R}^n$, which we call a $k$-piecewise constant vector. Since the best (in terms of $\ell_2$-norm) constant approximation to a function is its mean, a $k$-piecewise constant function approximation can be reparameterized over the collection of all disjoint intervals $\mathcal{I} = \{I_1, \ldots, I_k\}$ of $[n]$ such that given $\mathbf{x}$:

$$\min_{I_1, \ldots, I_k} \sum_{i=1}^{n} \sum_{j=1}^{k} (h_{I_j}(i) - x_i)^2 = \min_{I_1, \ldots, I_k} \sum_{j=1}^{k} \sum_{i \in I_j} \left( \frac{1}{|I_j|} \sum_{l \in I_j} x_l - x_i \right)^2 \tag{16}$$

We assume an optimal $H$ (parameterized by $\{I_i\}$ that can be obtained using many existing methods (a classical approach by dynamic programming [Jagadish et al., 1998]). The running time of such approaches is typically $O(nk)$, which is constant for fixed $k$; see Acharya et al. [2015] for a more detailed treatment.

Using Theorem 1, the Jacobian of the output $k$-histogram with respect to $\mathbf{x}$ assumes the following form:

$$\frac{\partial h}{\partial x_i} = \frac{\partial}{\partial x_i} \sum_{j=1}^{k} \left(\frac{1}{|I_j|} \sum_{l \in I_j} x_l)\right) = \frac{\partial}{\partial x_i} \sum_{j=1}^{k} \left(\frac{1}{|I_j|} (\sum_{l \in I_j} x_l) \mathbb{1}_{I_j}\right) \tag{17}$$

$$= \sum_{j=1}^{k} \frac{\partial}{\partial x_i} \frac{1}{|I_j|} (\sum_{l \in I_j} \mathbb{1}_{I_j}) = \frac{1}{|I_j|} \mathbb{1}_{I_j} \tag{18}$$

Therefore, the Jacobian of $h$ with respect to $\mathbf{x}$ forms the block-diagonal matrix $\mathbf{J} \in \mathbb{R}^{n \times n}$:

$$\mathbf{J} = \begin{bmatrix} \mathbf{J}_1 & \mathbf{0} & \dots & \mathbf{0} \\ \mathbf{0} & \mathbf{J}_2 & \dots & \mathbf{0} \\ \vdots & \vdots & \ddots & \vdots \\ \mathbf{0} & \mathbf{0} & \dots & \mathbf{J}_k \end{bmatrix}$$

where all entries of $\mathbf{J}_i \in \mathbb{R}^{|I_i| \times |I_i|}$ equal to $1/|I_i|$. Note here that the sparse structure of the Jacobian allows for fast computation, and it can be easily seen that computing the Jacobian vector product $\mathbf{J}^T \nu$ for any input $\nu$ requires $O(n)$ running time. As an additional benefit, the decoupling induced by the partition enables further speed up in computation via parallelization.

**Generalization to 1D piecewise polynomial fitting:** We now derive differentiable forms of generalized piecewise $d$-polynomial regression, which is used in applications such as spline fittings.

As before, $H : \mathbb{R} \to \mathbb{R}$ is any algorithm to compute the $k$-piecewise $d$ polynomial approximation of an input vector $\mathbf{x} \in \mathbb{R}^d$ that outputs partition $\mathcal{I} = \{I_1, \dots, I_k\}$. Similarly, the function $H$ gives us a vector $h \in \mathbb{R}^n$, a $k$-piecewise polynomial vector. Then, for each partition, we are required to solve a $d$-degree polynomial regressions. Generally, the polynomial regression problem is simplified to linear regression by leveraging a Vandermonde matrix. We get a similar closed-form expression for the coefficient as in Section 2.2.

Assume that for partition $I_j$, the input indices $t_{I_j}(i)$ is i$^{\text{th}}$ element in an index vector corresponding to the $I_j$ partition. Then, the input indices $t_{I_j}(i)$ are represented as a Vandermonde matrix, $\mathbf{V}_{I_j}$:

$$\mathbf{V}_{I_j} = \begin{bmatrix} 1 & t_{I_j}(1) & t_{I_j}(1)^2 & \cdots & t_{I_j}(1)^d \\ 1 & t_{I_j}(2) & t_{I_j}(2)^2 & \cdots & t_{I_j}(2)^d \\ \vdots & \vdots & \vdots & \ddots & \vdots \\ 1 & t_{I_j}(|I_j|) & t_{I_j}(|I_j|)^2 & \cdots & t_{I_j}(|I_j|)^d \end{bmatrix}.$$

It can be shown that the optimal polynomial coefficient $\alpha_{I_j}$ corresponding to the partition (or disjoint interval) $I_j$ have the following closed form:

$$\alpha_{I_j} = (\mathbf{V}_{I_j}^T \mathbf{V}_{I_j})^{-1} \mathbf{V}_{I_j}^T \mathbf{x}_{I_j},$$

where $\mathbf{x}_{I_j} \in \mathbb{R}^{|I_j|}$ is a vector $\mathbf{x}$ length of $|I_j|$ corresponding to the $I_j$ partition such that $\mathbf{x}_{I_j}(i) = x_i$ if $i \in I_j$ and undefined if $i \notin I_j$. This can be computed in $O(knd^w)$ time where $w$ is the matrix-multiplication exponent [Guha et al., 2006]. Then using Theorem 1 and the gradient for polynomial regression, the Jacobian of $h_{I_j}$ with respect to $\mathbf{x}$ forms a blockwise sparse matrix:

$$\frac{\partial h_{I_j}(s)}{\partial x_l} = \frac{\partial}{\partial x_l}(\langle \alpha_{I_j}, [\mathbf{V}_{I_j}^T]_s \rangle) = \frac{\partial}{\partial x_l}(\langle (\mathbf{V}_{I_j}^T \mathbf{V}_{I_j})^{-1} \mathbf{V}_{I_j}^T \mathbf{x}_{I_j}, [\mathbf{V}_{I_j}^T]_s \rangle)$$

$$= \frac{\partial}{\partial x_l} [\mathbf{V}_{I_j}^T]_s^T (\mathbf{V}_{I_j}^T \mathbf{V}_{I_j})^{-1} \mathbf{V}_{I_j}^T \mathbf{x}_{I_j}$$

$$= \begin{cases} \left[\mathbf{V}_{I_j}(\mathbf{V}_{I_j}^T \mathbf{V}_{I_j})^{-1}[\mathbf{V}_{I_j}^T])_s\right]_l & \text{if } l, s \in I_j \\ 0 & \text{otherwise.} \end{cases}$$

The two main takeaways here are as follows: (1) $\mathbf{V}_{I_i}$ can be precomputed for all possible $n-1$ partition sizes, thus allowing for fast ($O(n)$) computation of Jacobian-vector products; and (2) an added flexibility is that we can independently control the degree of the polynomial used in each of the partitions. The second advantage could be very useful for heterogeneous data as well as considering boundary cases in data streams.

### B.1 2D piecewise constant functions

Our 1D piecewise spline approximation can be (heuristically) extended to 2D data. We provide detailed descriptions. We consider the problem of image segmentation, which can be viewed as representing the domain of an image into a disjoint union of subsets. Neural-network-based segmentation involves training a model (deep or otherwise) to map the input image to a segmentation map, which is a piecewise constant spline function. However, standard neural models trained in a supervised manner with image-segmentation map pairs would generate pixel-wise predictions, leading to disconnected regions (or holes) as predictions. We leverage our approach to enforce deep models to predict piecewise constant segmentation maps. In case of 2D images, note that we do not have a standard primitive (for piecewise constant fitting) to serve as the forward pass. Instead, we leverage connected-component algorithms (such as Hoshen-Kopelman, or other, techniques [Wu et al., 2005]) to produce a partition, and the predicted output is a piecewise constant image with values representing the mean of input pixels in the corresponding piece. For the backward pass, we use a tensor generalization of the block Jacobian where each partition is now represented as a channel which is only non-zero in the positions corresponding to the channel. Formally, if the image $\mathbf{x} \in \mathbb{R}^n$ is represented as the union of $k$ partitions, $h = \bigcup_{i=1}^{k} I_i$, the Jacobian, $\mathbf{J_x} = \partial h / \partial \mathbf{x} \in \mathbb{R}^{n \times n}$ and,

$$\mathbf{J_x}(F(x))(s,t) = \begin{cases} \frac{\partial h(x)_s}{\partial x_t} = \frac{1}{|I_i|} & \text{if } s, t \in I_i, \\ 0 & \text{otherwise.} \end{cases} \tag{19}$$

Note that $I_i$ here no longer correspond to single blocks in the Jacobian. Here, they will reflect the positions of pixels associated with the various components. However, the Jacobian is still sparsely structured, enabling fast vector operations.

## C Implementing DSA with NURBS

### C.1 Backward evaluation for NURBS surface

In a modular machine learning system, each computational layer requires the gradient of a loss function with respect to the output tensor for the backward computation or the backpropagation. For our NURBS evaluation layer this corresponds to $\partial \mathcal{L}/\partial \mathcal{S}$. As an output to the backward pass, we need to provide $\partial \mathcal{L}/\partial \Psi$. While we represent $\mathcal{S}$ for the boundary surface, computationally, we only compute $\mathbf{S}$ (the set of surface points evaluated from $\mathcal{S}$). Therefore, we would be using the notation of $\partial \mathbf{S}$ instead of $\partial \mathcal{S}$ to represent the gradients with respect to the boundary surface. Here, we assume that with increasing the number of evaluated points, $\partial \mathbf{S}$ will asymptotically converge to $\partial \mathcal{S}$. Now, we explain the computation of $\partial \mathbf{S}/\partial \Psi$ in order to compute $\partial \mathcal{L}/\partial \Psi$ using the chain rule. To explain the implementation of the backward algorithm, we first explain the NURBS derivatives for a given surface point with respect to the different NURBS parameters.

### C.2 NURBS derivatives

We rewrite the NURBS formulation as follows:

$$\mathbf{S}(u,v) = \frac{\mathbf{NR}(u,v)}{w(u,v)} \tag{20}$$

where,

$$\mathbf{NR}(u,v) = \sum_{i=0}^{n} \sum_{j=0}^{m} N_i^p(u) N_j^q(v) w_{ij} \mathbf{P}_{ij}$$

$$w(u,v) = \sum_{i=0}^{n} \sum_{j=0}^{m} N_i^p(u) N_j^q(v) w_{ij}$$

For the forward evaluation of $\mathbf{S}(u,v) = \mathbf{f}(\mathbf{P}, \mathbf{U}, \mathbf{V}, \mathbf{W})$, we can define four derivatives for a given surface evaluation point: $\mathbf{S}_{,u} := \partial \mathbf{S}(u,v)/\partial u$, $\mathbf{S}_{,v} := \partial \mathbf{S}(u,v)/\partial v$, $\mathbf{S}_{,\mathbf{P}} := \partial \mathbf{S}(u,v)/\partial \mathbf{P}$, and $\mathbf{S}_{,\mathbf{W}} := \partial \mathbf{S}(u,v)/\partial \mathbf{w}$. Note that, $\mathbf{S}_{,\mathbf{P}}$ and $\mathbf{S}_{,\mathbf{W}}$ are represented as a vector of gradients $\{\mathbf{S}_{,P_{ij}} \forall P_{ij} \in \mathbf{P}\}$ and $\{\mathbf{S}_{w_{ij}} \forall w_{ij} \in \mathbf{W}\}$. Now, we show the mathematical form of each of these four derivatives. The first

derivative is traditionally known as the parametric surface derivative, $\mathbf{S}_{,u}$. Here, $N_{i,u}^p(u)$ refers to the derivative of basis functions with respect to $u$.

$$\mathbf{S}_{,u}(u,v) = \frac{\mathbf{NR}_{,u}(u,v)w(u,v) - \mathbf{NR}(u,v)w_{,u}(u,v)}{w(u,v)^2} \tag{21}$$

where,

$$\mathbf{NR}_{,u}(u,v) = \sum_{i=0}^{n}\sum_{j=0}^{m} N_{i,u}^p(u)N_j^q(v)w_{ij}\mathbf{P}_{ij}$$

$$w_{,u}(u,v) = \sum_{i=0}^{n}\sum_{j=0}^{m} N_{i,u}^p(u)N_j^q(v)w_{ij}$$

A similar surface point derivative could be defined for $\mathbf{S}_{,v}$. These derivatives are useful in the sense of differential geometry of NURBS for several CAD applications [Krishnamurthy et al., 2009]. However, since many deep learning applications such as surface fitting are not dependent on the $(u,v)$ parametric coordinates, we do not use them in our layer. Also, note that $\mathbf{S}_{,u}$ and $\mathbf{S}_{,v}$ are not the same as $\mathbf{S}_{,\mathbf{U}}$ and $\mathbf{S}_{,\mathbf{V}}$. A discussion about $\mathbf{S}_{,\mathbf{U}}$ and $\mathbf{S}_{,\mathbf{V}}$ is provided later in this section. Now, let us define $\mathbf{S}_{,p_{ij}}(u,v)$.

$$\mathbf{S}_{,\mathbf{P}_{ij}}(u,v) = \frac{N_i^p(u)N_j^q(v)w_{ij}}{\sum_{k=0}^{n}\sum_{l=0}^{m} N_k^p(u)N_l^q(v)w_{kl}} \tag{22}$$

$\mathbf{S}_{,\mathbf{P}_{ij}}(u,v)$ is the rational basis functions themselves. Computing $\mathbf{S}_{,w_{ij}}(u,v)$ is more involved with $w_{ij}$ terms in both the numerator and the denominator of the evaluation.

$$\mathbf{S}_{,w_{ij}}(u,v) = \frac{\mathbf{NR}_{,w_{ij}}(u,v)w(u,v) - \mathbf{NR}(u,v)w_{,w_{ij}}(u,v)}{w(u,v)^2} \tag{23}$$

where,

$$\mathbf{NR}_{,w_{ij}}(u,v) = N_i^p(u)N_j^q(v)\mathbf{P}_{ij}$$

$$w_{,w_{ij}}(u,v) = N_i^p(u)N_j^q(v)$$

### C.3 Derivatives with respect to knot points

For simplicity, we will stick to 1D NURBS curves. The extension to 2D surfaces is straightforward using Kronecker products.

We recall the definition of the NURBS basis:

$$N_i^d(u) = \frac{u - u_i}{u_{i+d} - u_i}N_i^{d-1}(u) + \frac{u_{i+d+1} - u}{u_{i+d+1} - u_{i+1}}N_{i+1}^{d-1}(u), \ N_i^0(u) = \left\{ \begin{array}{ll} 1 & \text{if } u_i \leq u \leq u_{i+1} \\ 0 & \text{otherwise} \end{array} \right. \tag{24}$$

The goal is to evaluate the derivative of $N_i^d(u)$ with respect to the knot points $\{u_i\}$. We observe that due to the recursive nature of the definition, we can accordingly compute the derivatives of $N_i^d(u)$ in a recursive fashion using the chain rule, *provided* we can evaluate:

$$\frac{\partial N_i^0(u)}{\partial u_i} = \frac{\partial \mathbf{1}([u_i, u_{i+1}])}{\partial u_i}$$

(and likewise for $u_{i+1}$) where $\mathbf{1}$ denotes the indicator function over an interval. However, this derivative is not well-defined since the gradient is zero everywhere and undefined at the interval edges.

We propose to approximate this derivative using *Gaussian smoothing*. Rewrite the interval as the difference between step-functions convolved with deltas shifted by $u_i$ and $u_{i+1}$ respectively:

$$\mathbf{1}([u_i, u_{i+1}))(u) = \text{sign}(u) \star \delta(u - u_i) - \text{sign}(u) \star \delta(u - u_{i+1})$$

and approximate the delta function with a Gaussian of sufficiently small (but constant) bandwidth:

$$\mathbf{1}([u_i, u_{i+1}])(u) = \text{sign}(u) \star G_\sigma(u - u_i) - \text{sign}(u) \star G_\sigma(u - u_{i+1})$$

where

$$G_\sigma(u - \mu) = \frac{1}{\sqrt{2\pi\sigma^2}} \exp(-\frac{(u - \mu)^2}{2\sigma^2}).$$

The derivative with respect to $\mu$ is therefore given by:

$$G_\sigma'(u = \mu) = \frac{(u - \mu)}{2\sigma^2} G_\sigma(u - \mu),$$

which means that the approximate gradient introduces a multiplicative $(u - \mu)$ factor with the original basis function. Propagating this through the chain rule and applying a similar strategy as Cox-de Boor recursion gives us Algorithm 1. □

## D  Experimental details

### D.1  Segmentation

**Weizmann Horse dataset:**  The dataset consists of 378 images of single horses with varied backgrounds and their corresponding ground truth. We divide the dataset into 85:15 ratios for training and testing, respectively. Further, each image is normalized to a $[0, 1]$ domain by dividing it by 256. 5443

**Cell dataset:**  The dataset consists of 19K gray-scale images containing various cells, and we take 1900 subset images as the dataset. We divide the dataset into 85:15 ratios for training and testing, respectively. Similarly, we normalize the image to a $[0, 1]$ by dividing each pixel by 256.

**Architecture and training:**  We use the following U-Net architecture for training our segmentation networks. While we use the equivalent model skeleton reported by Ronneberger et al. [2015], we scale down the network size starting the initial channels $C = 8$ (default channel is $C = 64$). In both datasets, we train the network 1000 epochs with an initial learning rate of 0.0003. We leverage Adam optimizer with $\beta = (0.9, 0.999)$ and weight decay 0.0001. We use a binary cross-entropy loss function as the objective function.

### D.2  NURBS surface fitting implementation

The complete algorithm for forward evaluation of $\mathbf{S}(u, v)$ as described in Piegl and Tiller [1997] can be divided into three steps:

1. Finding the knot span of $u \in [u_i, u_{i+1})$ and the knot span of $v \in [v_j, v_{j+1})$, where $u_i, u_{i+1} \in \mathbf{U}$ and $v_j, v_{j+1} \in \mathbf{V}$. This is required for the efficient computation of only the non-zero basis functions.

2. Now, we compute the non-zero basis functions $N_i^p(u)$ and $N_j^q(v)$ using the knot span. The basis functions have specific mathematical properties that help us in evaluating them efficiently. The partition of unity and the recursion formula ensures that the basis functions are non-zero only over a finite span of $p + 1$ control points. Therefore, we only compute those $p + 1$ non-zero basis functions instead of the entire $n$ basis function. Similarly in the $v$ direction we only compute $q + 1$ basis functions instead of $m$.

3. We first compute the weighted control points $\mathbf{P}_{ij}^w$ for a given control point $\mathbf{P}_{ij} = \{\mathbf{P}_x, \mathbf{P}_y, \mathbf{P}_z\}$ and weight $w_{ij}$ as $\{\mathbf{P}_x w, \mathbf{P}_y w, \mathbf{P}_z w\}$ representing the surface after homogeneous transformation for ease of computation. Once the basis functions are computed we multiply the non-zero basis functions with the corresponding weighted control points, $\mathbf{P}_{ij}^w$. This result, $\mathbf{S}'$ is then used to compute $\mathbf{S}(u, v)$ as $\{S_x'/S_w', S_y'/S_w', S_z'/S_w'\}$.

---

**Algorithm 2** Forward algorithm for multiple surfaces

---

**Input** : $\mathbf{U}$, $\mathbf{V}$, $\mathbf{P}$, $\mathbf{W}$, output resolution $n_{grid}$, $m_{grid}$
**Output** : $\mathbf{S}$

Initialize a meshgrid of parametric coordinates
      uniformly from $[0, 1]$ using $n_{grid} \times m_{grid} : u_{grid} \times v_{grid}$
Initialize: $\mathbf{S} \rightarrow \mathbf{0}$
**for** $k = 1 : surfaces$ **in parallel do**
    **for** $j = 1 : m_{grid}$ *points* **in parallel do**
        **for** $i = 1 : n_{grid}$ *points* **in parallel do**
            Compute $u_{span}$ and $v_{span}$ for the corresponding $u_i$ and $v_i$ using knot vectors $\mathbf{U_k}$ and $\mathbf{V_k}$
            Compute basis functions $N_i$ and $N_j$ basis functions using $u_{span}$ and $v_{span}$ and knot vectors $\mathbf{U_k}$ and $\mathbf{V_k}$
            Compute surface point $\mathbf{S}(u_i, v_j)$ (in $x$, $y$, and $z$ directions).
            Store $u_{span}$, $v_{span}$, $N_i$, $N_j$, and $\mathbf{S}(u_i, v_j)$ for backward computation

---

In a deep learning system, each layer is considered as an independent unit performing the computation. The layer takes a batch of input during the forward pass and transforms them using the parameters of the layer. Further, in order to reduce the computations needed during the backward pass, we store extra information for computing the gradients during the forward computation. The NURBS layer takes as input the control points, weights, and knot vectors for a batch of NURBS surfaces. We define a parameter to control the number of points evaluated from the NURBS surface. We define a mesh grid of a uniformly spaced set of parametric coordinates $u_{grid} \times v_{grid}$. We perform a parallel evaluation of each surface point $S(u, v)$ in the $u_{grid} \times v_{grid}$ for all surfaces in the batch and store all the required information for the backward computation. The complete algorithm is explained in Algorithm 2. Our implementation is robust and modular for different applications. For example, if an end-user desires to use this for a B-spline evaluation, they need to set the knot vectors to be uniform and weights $\mathbf{W}$ to be $1.0$. In this case, the forward evaluation can be simplified to $\mathbf{S}(u, v) = \mathbf{f}(\mathbf{P})$. Further, we can also pre-compute the knot spans and basis functions during the initialization of the NURBS layer. During computation, we could make use of tensor comprehension that significantly increases the computational speed. We can also handle NUBS (Non-Uniform B-splines), where the knot vectors are still non-uniform, but the weights $W$ are set to $1.0$. Note in the case of B-splines $\Psi = \{\mathbf{P}\}$ (the output from the deep learning framework) and in the case of NUBS $\Psi = \{\mathbf{P}, \mathbf{U}, \mathbf{V}\}$.

**SplineNet training details:** The SplineNet architecture comprises a series of dynamic graph convolution layers, followed by an adaptive max pooling and conv1d layers. We use the Chamfer distance as the loss function. The Chamfer distance ($\mathcal{L}_{CD}$) is a global distance metric between two sets of points, as shown below.

$$\mathcal{L}_{CD} = \sum_{\mathbf{P_i} \in \mathbf{P}} \min_{\mathbf{Q_j} \in \mathbf{Q}} ||\mathbf{P_i} - \mathbf{Q_j}||_2 + \sum_{\mathbf{Q_j} \in \mathbf{Q}} \min_{\mathbf{P_i} \in \mathbf{P}} ||\mathbf{P_i} - \mathbf{Q_j}||_2 \tag{25}$$

For training and testing our experiments, we use the SplineDataset provided by Sharma et al. [2020]. The SplineDataset is a diverse collection of open and closed splines that have been extracted from one million CAD geometries included in the ABC dataset. We run our experiments on open splines split into 3.2K, 3K, and 3K surfaces for training, testing, and validation.

### D.3 PDE solver implementation with DSA prior

Deep convolutional neural networks are a natural choice for the network architecture for solving PDEs due to the structured grid representation of $\mathcal{S}^d$ and similarly structured representation of $U_\theta^d$. The spatial localization of convolutional neural networks helps in learning the interaction between the discrete points locally. Since the network takes an input of a discrete grid representation (similar to an image, possibly with multiple channels) and predicts an output of the solution field of a discrete grid representation (similar to an image, possibly with multiple channels), this is considered to be similar to an image segmentation or image-to-image translation task in computer vision. U-Nets [Ronneberger et al., 2015] have been known to be effective for applications such as semantic segmentation and image reconstruction.

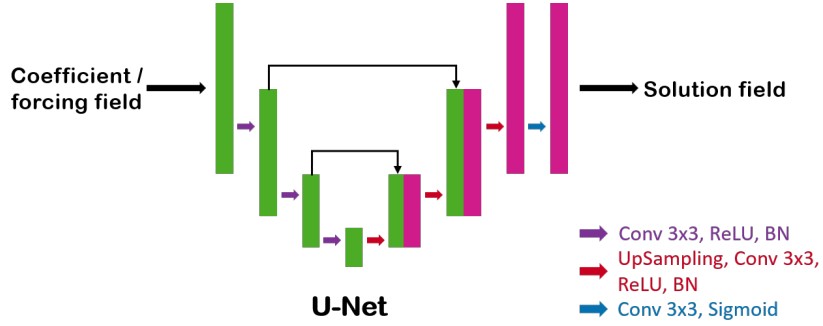

Figure 4: UNet architecture used for training

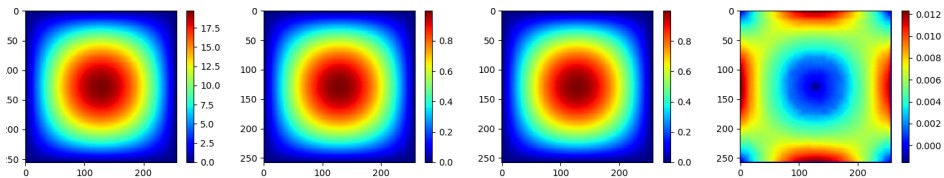

Figure 5: Solution to the linear Poisson's equation with forcing. From left to right: $f$, $u_{DSA}$, $u_{num}$ and $(u_{DSA} - u_{num})$. Here $u_{num}$ is a conventional numerical solution obtained through FEM. Diffusivity $\nu = 1$

We choose U-Net architecture for solving the PDE due to its success in other diverse applications. The architecture of the network is shown in Figure 4. First, a block of convolution and instance normalization is applied. Then, the output is saved for later use during skip-connection. This intermediate output is then downsampled to a lower resolution for a subsequent convolution block and instance normalization layers. This process is continued twice. The upsampling starts where the saved outputs of similar dimensions are concatenated with the output of upsampling for creating the skip-connections followed by a convolution layer. LeakyReLU activation was used for all the intermediate layers. The final layer has a Sigmoid activation.

### D.3.1 Applying boundary conditions

The Dirichlet boundary conditions are applied exactly. The query result from $U_\theta^d$ from the network pertains only to the interior of the domain. The boundary conditions need to be taken into account separately. There are two ways of doing this:

- Applying the boundary conditions exactly (this is possible only for Dirichlet conditions in FEM/FDM, and the zero-Neumann case in FEM)

- Taking the boundary conditions into account in the loss function, thereby applying them approximately.

We take the first approach of applying the Dirichlet conditions exactly (subject to the mesh). Since the network architecture is well suited for 2d and 3d matrices (which serve as an adequate representation of the discrete field in 2D/3D on regular geometry), the imposition of Dirichlet boundary conditions amounts to simply padding the matrix by the appropriate values. A zero-Neumann condition can be imposed by taking the "edge values" of the interior and copying them as padding. A nonzero Neumann condition is slightly more involved in the FDM case since additional equations need to be constructed, but if using FEM loss, this can be done with another surface integration on the relevant boundary.

# E   Additional results

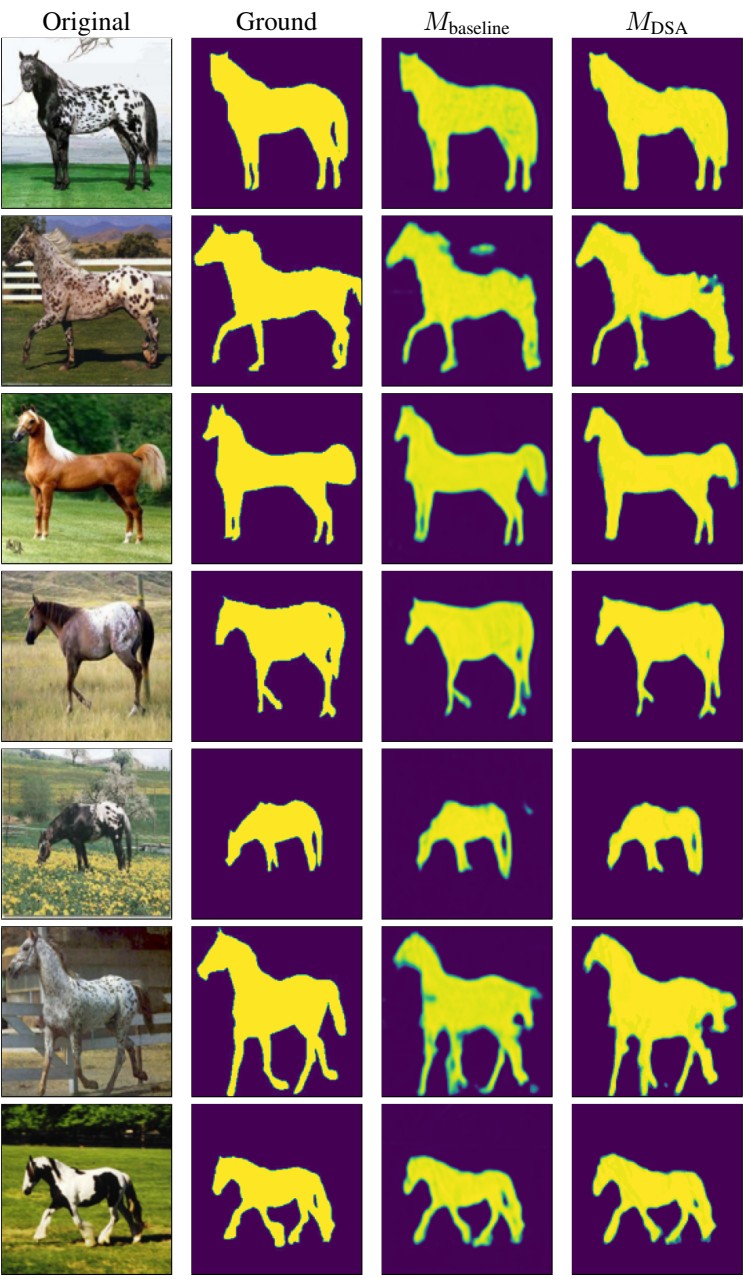

Figure 6: **Image Segmentation Tasks** Adding DSA layers ($M_{\text{DSA}}$) on top of U-Net ($M_{\text{baseline}}$) improves the segmentation tasks on both datasets.

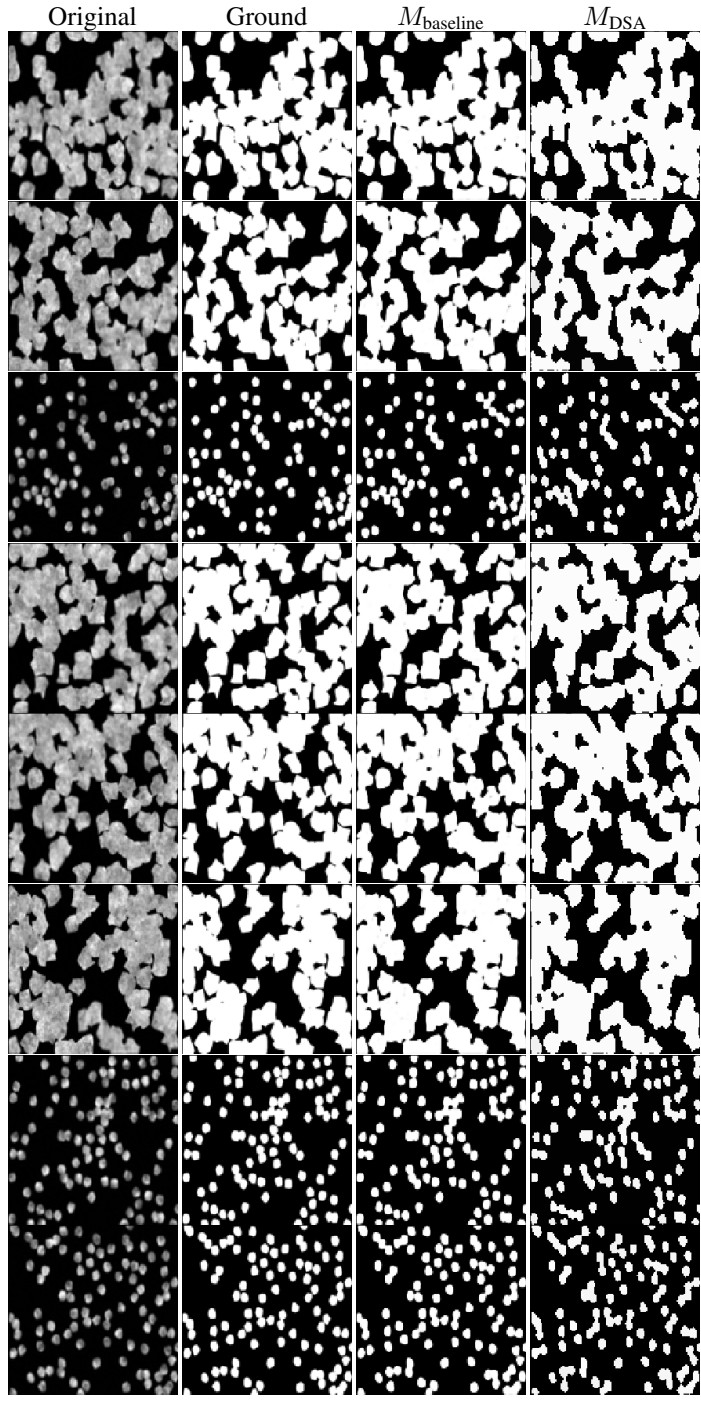

Figure 7: Additional cell segmentations results. $M_{\text{baseline}}$ and $M_{\text{DSA}}$ correspond to U-Net and U-Net+DSA layers, respectively.

**Ablation studies:** In the main paper, we demonstrate how our module can perform for different experiments. In this section, we assess the computational performance of our module. For brevity, we restrict our analysis to surface fitting operation and analyze the timings with variations in the number of control points, evaluation points, and surface degree. We only study the first 500 iterations (which include both the forward and backward pass). We perform all our experiments on a desktop with a 32 core 2.4 GHz Intel Xeon processor, 64 GB RAM, and an NVIDIA Titan Black GPU with 6 GB RAM.

Table 6: Time to fit a surface for different number of control points.

| Control Points | Iteration time (s) |
|---|---|
| $6 \times 6$ | 0.098 |
| $12 \times 12$ | 0.106 |
| $24 \times 24$ | 0.110 |
| $48 \times 48$ | 0.110 |

Table 7: Time to fit a surface for different number of evaluation points.

| Evaluation Points | Iteration time (s) |
|---|---|
| $64 \times 64$ | 0.074 |
| $128 \times 128$ | 0.120 |
| $256 \times 256$ | 0.170 |
| $512 \times 512$ | 0.266 |

Table 8: Computation time to fit a surface of different degree.

| Degree | Iteration time (s) |
|---|---|
| 1 | 0.074 |
| 2 | 0.120 |
| 3 | 0.170 |
| 4 | 0.266 |