# OpenReview forum: "Differentiable Spline Approximations"
_NeurIPS.cc/2021/Conference — NeurIPS 2021 Poster_

### Official Review · Reviewer_Q18T · 2021-07-12

**Rating:** 7
**Confidence:** 4

**Summary:**

The paper presents sparse Jacobians/backward algorithms for several spline formulations and motivates their use in differentiable algorithms. The formulations include general splines, NURBS, and basis functions for FEM solvers.
The resulting differentiable modules are applied to the three proof of concept tasks of image segmentation, point cloud reconstruction, and solving Poisson equations, where they show promising results.

**Limitations And Societal Impact:**

The authors could further discuss the limitations of their method. This would help to assess the applicability of the presented approach for real-world tasks. I see no immediate societal impact.

**Main Review:**

Originality:
- To my knowledge, this is the first method that allows backpropagating gradients to the knot vectors of NURBS curves. The authors achieve this by framing the box function as convolutions with Dirac impulses and approximating those Dirac impulses using Gaussian functions. This seems to be an interesting idea, which possibly could be applied to other situations as well.
- The observation that the same principles can be applied to backpropagate through functions in FEM solvers is interesting and has potential. There might be several useful applications.
- Utilizing the sparsity of the Jacobian (induced by the local support property) to develop efficient backward algorithms through splines was already done by previous work (e.g. SplineCNN [1]). However, this is the first work that presents that concept in a principled way for general splines.

Quality:
- The paper is technically sound
- The experiments show that the method is indeed applicable in several domains.
- All three experiments are proofs of concept. I wonder what hinders the application to more sophisticated tasks. At least for image segmentation it should be possible to compare to SOTA methods on common segmentation benchmarks. Instead, the authors only compare to their own baseline.

Clarity:
- The paper is well-written, well structured and nice to read.
- The paper would profit from some explanatory figures for the general splines in Section 2/2.1 and the FEM method in Section 2.3.

Significance:
- This is a paper which can spark a lot of applications in several areas, if the presented methods turn out to be practical. There are several interesting ideas and concepts.

All in all, I recommend to accept the paper for the mentioned reasons. The experiments certainly leave room for improvement, mostly because the authors are not comparing to other SOTA methods. However, the given experiments reach the bar in that they showcase the different potential applications as proofs of concept.

Questions:
- Is the approximation of Basis functions (using Gaussians instead of Dirac impulses) only done in the backward step or also in the forward step? If it is done in both I wonder if the approach would also work if the normal basis is used in the forward computation, while the "surrogate" is used for gradient computation.
- How efficient is the FEM solver? Would it be possible to apply it to larger scale tasks, for example volumetric reconstruction tasks as in SIREN [2]?
- In the given applications, the knot vectors (and therefore the gradients computed for them) lie in 2- or 3-dimensional space. When performing gradient descent in such low dimensional spaces, one often runs into bad local minima. Did the authors face any of such challenges when optimizing their positions?


[1] Fey et al.: SplineCNN: Fast Geometric Deep Learning with Continuous B-Spline Kernels, CVPR 2018

[2] Sitzmann et al.: Implicit Neural Representations with Periodic Activation Functions, NeurIPS 2020

----------------- After Rebuttal ---------------

I thank the authors for answering my questions. I will keep my score.

The other reviewers correctly note that in this paper some contributions are oversold, as both ideas, utilizing sparsity of the Jacobian for efficient backprop and computing gradients through Dirac functions via smooth approximations are not novel but already present in previous work. I agree and encourage the authors to adjust the statements in future versions.

However, all in all, I see a lot of value in this work, as it is the first that builds a full differentiable building block for general splines, which can be useful to many researchers.

**Time Spent Reviewing:**

6

---

> ### Author Response · Authors · 2021-08-10
> **Response**
>
> We thank the reviewer for their positive feedback. Here are the specific answers to the questions:
>
> 1. Thanks for the insightful comment. Indeed, the Gaussian-convolved approximation is used only while deriving the gradients in the backward pass (Algorithm 1). The forward step uses the standard basis function evaluation computed recursively using the Cox-de Boor algorithm.
> 2. The FEM solver is very effective if we define a Galerkin least-squares formulation for the chosen PDE. There are some differences between the approach taken in SIREN and our work. SIREN uses *implicitly* defined neural networks to solve the PDE (or perform volumetric reconstruction), while in our approach, we use splines to approximate differential operators in the loss computation. Since the operations are performed per mesh element, we have efficiently implemented this using a set of 3D convolution operations, which makes our approach basically as efficient as 3D CNNs that perform video classification or volumetric segmentation.
> 3. Thanks for the insightful comment. Indeed, training over knot parameters is challenging, as rightly pointed out by the reviewer, and proper choice of bandwidth (seen in the additional results table provided above in the response to Review 2) seems crucial. Other than bandwidth tuning, we use two different optimization loops (akin to alternating minimization). Optimizing for control points and weights is much easier than optimizing for the knot vectors (since the problem is linear); this difference in the training dynamics between the knot vectors and the control points could lead to drastic “mode collapse”: the intermediate knots could merge to a common knot, making it difficult to recover from such a condition (since repeated knots leads to discontinuity in the spline basis functions). Therefore, we first run a few gradient steps for control points/weights and then run a single gradient update for the knot vector. Such an alternating minimization algorithm helps us to stabilize the gradient updates.
> 4. Regarding the above comment on SOTA comparisons: we do compare our results for the surface reconstruction task with (SOTA) SplineCNN and show improvements. Also, we compare the results of our PDE solver with PINNs (one of the SOTA methods for solving PDEs in a data-driven manner) and achieve improvements.
>
> We would like to emphasize (as rightly pointed out by the reviewer) that our goal in this paper was not necessarily to reach SOTA in every application. Indeed, for standard computer vision problems such as pixel-level image segmentation, highly optimized and hand-crafted deep architectures already exist (along with the massive datasets, such as COCO, required to train them). It is not unreasonable to expect that any SOTA improvements may be incremental at this point. Our goal instead is to fill the gap between the rich literature on spline fitting from the geometric modeling community, and current trends in modern ML that seem to focus on differentiable learning.

---

### Official Review · Reviewer_ZLi8 · 2021-07-12

**Rating:** 6
**Confidence:** 3

**Summary:**

The paper presented a differentiable approximation to splines for use as a differentiable layer in differential programming. The main idea is to exploit the sparsity of the resulting Jacobian matrix for the spline approximation operation, and to use a Guassian approximation for the non-differentiable Direc delta spike functions from the Cox-de Boor base function. The authors used this differentiable spline approximation in three application settings: 2d image segmentation; 3d surface fitting for point clouds using NURBS, and solving a PDE (Poisson Equation).

**Limitations And Societal Impact:**

In general I think differentiable spline is an interesting topic. A few areas of improvement I would love to see:
1. More comprehensive ablations on the smoothing part for differentiability.
2. Clear details for the experiments. If space is an issue, consider moving more details to the suppmat.
3. Some performance benchmarking would be nice-to-have. How does the algorithm performance scale with number of knots / degree? How fast is this layer compared with other NN layers?

**Main Review:**

The paper is rather math dense and it took me some time to understand the main proposal set forth by this work. The main pros and cons are concluded below.

Pros:
+ I think the differentiable approximation of the spline functions is a very interesting topic, potentially allowing the integration of spline function approximations into the differentiable programming paradigm. The current field of deep learning is heavily dominated by "neural" (i.e., Multi-Layer Perception) approximations for parameterizing high dimensional functions, which are flexible and extensible, but has its own set of limitations. I think generalizing the idea of differentiable splines as function approximations is attractive.
+ The idea of smoothing the discontinuous delta functions using Gaussian with small bandwidth feels intuitive.
+ I like the error intervals reported in the experiments.

Cons:
- Though the paper is quite content heavy, most of the stuff is not new (e.g., when computing the Jacobian, only the gradients corresponding to the specific segments are non-zero). The main contribution seems to be that the authors proposed to smooth the step function by approximating it with a Gaussian kernel. However, this main aspect does not seem very thoroughly studied.
    - For example, how does the bandwidth of the Gaussian affect the results (with empirical evidence)? I would assume a bandwidth too small will result in nasty gradients, and a bandwidth too large will produce inaccurate approximations.
    - What if you replace the step function in the base case (Eqn. 4) with two sigmoids of different bandwidth instead of using a Gaussian? What about using two tanh? Different options can be explored in greater depth.
- The experimental section is not very well executed.
    -  Following the point above, ablation studies for the choice of smoothing function are lacking.
    - The experiments lack details and context. For example, I find it difficult to understand how the differentiable spline layer is used in the image segmentation experiments. The only description seems to be "We begin with implementing a 2D piecewise constant splines regression approach for the image segmentation problem using a UNet" (l 242). What does that mean? Are the shape contours represented using NURBS? How is the output parameterized? What parameters are the neural network outputting after the UNet feature map? The baseline method is simply described as "without the DSA layer", with not enough details about the DSA layer for the reader to understand. The evaluation is a little hard to grasp. How is the Jaccard score defined? Is it the same as pixel-wise IoU score? For the PDE experiment, is my understanding correct that the first part looks at directly optimizing the Lagrange polynomial based representation (by minimizing PDE residuals), and the second part looks at training for a learned prior over a dataset (using a UNet)? Similar question to the first experiment, how are the U-Net feature maps associated with the spline polynomials?

**Time Spent Reviewing:**

3 hours

---

> ### Author Response · Authors · 2021-08-10
> **Response**
>
> We thank the reviewer for their positive feedback.
>
> We agree that the presentation of our results can (and should) be improved. We provide a table below summarizing the three applications presented in our paper, the corresponding input data, network output, model architecture, the choice of basis functions used for splines, and the degree of the splines. We hope this clarifies some of the confusion and we will add these details in an additional section, as helpfully suggested by the reviewer.
>
> | Application  | Input  | Output  | Model Architecture  | Spline Basis Functions |Degree of Splines |
> |---|---|---|---|---|---|
> |Image Segmentation   | Image  | 2D piecewise constant knot partitions  | U-Net   | Box-Car functions  | 0 |
> |Point cloud reconstruction| 3D point cloud | 3D control points and rational weights  | Dynamic Graph CNN  |B-Spline polynomial functions  | 3 |
> | PDE based surrogate physics  | 2D Mesh (regular grid)  |  2D physics field value on the mesh grid | U-Net | Lagrange polynomial functions | 1,2,3|
>
> Additional details:
> 1. Ablation studies with different Gaussian bandwidths
>
> First, we report ablation studies that show that our choice of bandwidth hyperparameter over a wide range leads to good results. Consider the surface fitting experiment. Since we cannot provide figures in this response, instead we provide training trends after 500 iterations. We notice that most “small” bandwidth parameter choices work. (For very, very small choices of $\sigma$ we run into overflow issues).  As we keep increasing the bandwidth, the gradients become too noisy and begin to diverge (as shown in the Table below).
>
> | Experiment | Behavior after 500 iterations   |
> | --- | --- |
> |Gaussian width ($\sigma=0.0001$) | diverges |
> |Gaussian width ($\sigma=0.001$) | converges |
> |Gaussian width ($\sigma=0.01$) | converges (eventually leads to best results) |
> |Gaussian width ($\sigma=0.1$) | converges |
> |Gaussian width ($\sigma=0.5$) | begins to diverge |
> |Gaussian width ($\sigma=1.0$) | diverges |
>
> 2. Experiments with different smoothing functions
>
> Thanks for the suggestion, and we agree that there are other smooth alternatives to the hard step function (such as sigmoids). However, because of the following 3 reasons, our approach leads to a naturally recursive algorithm to compute the derivative (of the smoothed spline). (i) B-splines are recursively defined using boxcar functions (which are differences of hard steps), (II) Gaussian convolutions commute nicely with summations, and (iii) convolutions are easy to implement using modern deep learning packages. However, we do agree that identifying the “best” smoother is a very interesting theoretical question.
>
> 3. Timings with number of knots/number of control points
>
> We thank the reviewer for the insightful comment. We will add the following table to the supplementary material. We begin with the minimum number of knots for which we would get meaningful surfaces for the given experiment (i.e. 6x6 control points). By modifying the number of control points, we can directly modify the number of knots. For benchmarking the timings and performance we only study the first 500 iterations of the optimization (which includes both the forward and backward pass).  All these experiments are run on a PC (32 core 2.4GHz Intel Xeon processor with 64 GB RAM) with CPU support only.
>
> | | $6 \times 6$ | $12 \times 12$ | $24 \times 24$ | $48 \times 48$ |
> | --- | --- | --- | --- | --- |
> |Iteration time (s) | 0.098| 0.106| 0.110| 0.110 |
> |MSE | 22719.6096 | 8.7109 | 0.0728 | 0.0000 |
>
> 4. Timings with number of evaluated points
>
> We wanted to point out that the key increase in the computational effort is also affected by the number of evaluated points which we show in the table below. We will add this table to the supplementary material. Same as before, for benchmarking the timings and performance, we only study the first 500 iterations of the optimization (which includes both the forward and backward passes). All these experiments are run on a PC (32 core 2.4GHz Intel Xeon processor with 64 GB RAM) with CPU support only.
>
> | |  $64 \times 64$ | $128 \times 128$ | $256 \times 256$ | $512 \times 512$ |
> | --- | --- | --- | --- | --- |
> |Iteration time (s) | 0.116 | 0.120 | 0.136 | 0.194|
> |MSE | 811.248 | 721.712 | 701.221 | 637.482 |
>
> 5. Timings with different degrees of the splines
>
> We perform a similar study for the different spline degrees.
>
> | |  $d=1$ | $d=2$ | $d=3$ | $d=4$ |
> | --- | --- | --- | --- | --- |
> |Iteration time (s) | 0.074 | 0.120| 0.170| 0.266|
> |MSE | 260.649 | 55.555 | 69.702| 83.330 |
>
> *Specific details for Image Segmentation*:
>
> We would like to clarify our experimental setup for the image segmentation experiments. In line 242, the "2D piecewise constant splines regression approach" refers to a discretized spline fitting with 0-degree basis functions (box-car functions).
>
> We use the equivalent U-Net architecture to Ronneberger et al. but scale down the network size starting with the initial channels $C=8$. (Line 640 in Appendix). Thus, the U-Net architecture we used in the segmentation experiment has an output dimension that is the same as the input dimension (which is typical in image-segmentation).
>
> On top of the U-Net architecture, we append the DSA layer, which forces the output to be piecewise constant. The forward pass involves partitioning the output images by a standard connected-components algorithm to produce the partition and outputs a piecewise constant image with values representing the mean of input pixels in the corresponding pieces of the partition (Line 574). The partition computed in the forward pass is used to define the block-sparse Jacobian during the backward pass.
>
> We use the standard definition of the Jaccard score, which, as you correctly point out,  is the same as the pixel-wise IoU score.
>
> We plan to add these details to the supplement.

---

### Official Review · Reviewer_qxvV · 2021-07-16

**Rating:** 5
**Confidence:** 4

**Summary:**

The authors propose to derive a technique allowing gradient based learning of splines (or piecewise polynomials) and to apply such a method to standard tasks such as PDE solving/approximation and other function approximation problems.

**Limitations And Societal Impact:**

yes

**Main Review:**

Some notations are counter intuitive e.g. f_I(i) since F is a vector and F_I is a restriction (in term of dimension) of this version, it seems unnatural to call it as a function (especially since it is not well introduced/defined).

The problem of both optimizing the spline (per-region) function parameters and the knots of the partition is well studied yet no reference is provided (see all the literature on adaptive spline, and many DeVore references).

Theorem1 is merely a direct calculus result that is straightforward and has been known in the spline community for a while (it is trivial to show the block structure based on the fact that each subset of parameters is only ‘’active’’ within its partition region)

Many english formulation that are not appropriate ‘’the backward pass is a bit more tricky’’ and so on. Additionally the whole organization and choice of organization is odd (e.g. bold paragraph in the intro and so on).

Many sentences seem incomplete e.g. “This is powerful since ….into equivalent energy functionals”.

No comparison is done against alternative spline fitting methods including the ones that allow adaptivity of the partition though mesh refinement (which is the whole direction of PDEs where mesh refinement is able to somewhat compete with adaptive knot splines).


----- AFTER REBUTTAL ------

I thank the authors for their detailed answers and acknowledging the drawbacks I have raised. I am raising my score from 4 to 5 for the following reason. In my opinion, some of the technical contributions that are the pillars of the paper have been obtained before (e.g. using Gaussian (or more generally narrow support function) approximation for gradient computation in https://arxiv.org/pdf/1706.04698.pdf https://arxiv.org/pdf/1901.09948.pdf, gradient sparsity which is (as also pointed out by ZLi8) known and already leveraged in various fields such as PDE approximations (the sparsity of the gradient is what allows most methods to actually be tractable on a fine mesh)). And when considering the remaining contributions, I am uncertain that this paper passes the acceptance threshold. However, some reviewers are pointing out that this paper offers an interesting application and find the submission useful, hence I am raising my score accordingly.

**Time Spent Reviewing:**

5

---

> ### Author Response · Authors · 2021-08-10
> **Response**
>
> We thank the reviewer for their constructive criticism. First, in terms of minor points, we will correct all inappropriate English formulations and incomplete sentences. We will also restrict the use of the word “function” since all our quantities reside in finite-dimensional vector spaces.
>
> We acknowledge the rich literature on adaptive spline approximations, dating back to the foundational work by DeVore and co-authors in the late eighties and nineties. We will be happy to include many such references. Indeed, fitting 1D splines is effectively a solved problem in approximation theory. However, to our knowledge, ours is the first work to perform *gradient-based* optimization of knots, control points, and other spline parameters. We emphasize that *gradient-based* spline approximation is *particularly relevant in the context of modern machine learning* models (such as deep neural networks), whose very essence is the recursive composition of *differentiable modules* that support gradient-based learning. If there are any other papers to which the reviewer can point us, which pursue gradient-based spline fitting, we will be glad to properly acknowledge them.
>
> While the proof of Theorem 1 is easy, we are not aware if the result is well-known. If the reviewer can point us to any concrete references in the spline literature which makes a formal observation along these lines, we will be grateful. Note also that similar results involving block-sparse Jacobians have very recently appeared in the theoretical machine learning literature on implicit differentiation; see [2] and [3].
>
> Further, we would like to point out that we are not performing knot refinement, or adaptive partitioning and mesh refinement, since that is not directly our focus. Instead, our focus is a *differentiable* framework for approximation with spline-like functions (including non-uniform rational B-splines), as correctly summarized by Reviewer #3. An interesting open question along this direction is whether knot refinement and adaptive spline fitting can be differentiably performed; in both cases, we would be adding additional control points or knots; this is a non-differentiable operation and it will be interesting to derive a formal differentiable approximation. Similarly, knot reparameterization (which we do address in our method) is also not a differentiable operation. Note that knot reparameterization is still a major challenge in geometric modeling, and recent approaches to perform knot reparameterization of Bezier curves use deep learning for this task, but require massive amounts of training data [1].
>
> Geometric modeling, segmentation, and PDE solvers are three among many other applications where spline approximations arise (and due to limitations of time and space, we only include three “showcase” applications). We hope that the reviewer will evaluate our work in a broader context: we are not merely after a better surface fitting method or better PDE solver, but rather our goal is a general computational tool for learning-friendly “differentiable” spline fitting (of which Surface Reconstruction is only an example task). Future follow-up work could include other applications, exploring other types of geometric priors (constraints such as higher order continuity), and further numerical quantification of the inductive bias of spline-type priors in deep neural architectures.
>
> [1] Scholz, Felix, and Bert Jüttler. "Parameterization for polynomial curve approximation via residual deep neural networks." Computer Aided Geometric Design 85 (2021): 101977.
>
> [2] Bertrand, Q., Klopfenstein, Q., Blondel, M., Vaiter, S., Gramfort, A. and Salmon, J.,. Implicit differentiation of Lasso-type models for hyperparameter optimization. In International Conference on Machine Learning, 2020.
>
> [3] Bertrand, Q., Klopfenstein, Q., Massias, M., Blondel, M., Vaiter, S., Gramfort, A., & Salmon, J. (2021). Implicit differentiation for fast hyperparameter selection in non-smooth convex learning. arXiv preprint arXiv:2105.01637.

---

### Author Response · Authors · 2021-08-26
**Response to reviewers**

Dear Reviewers,

Thank you very much for the thoughtful reviews and constructive criticism. We hope that we have satisfactorily addressed your concerns in our point-by-point responses below. Since the discussion period is ending soon, we would be grateful if you could let us know if you have any follow-up questions.

-Authors

---

### Decision · Program_Chairs · 2021-09-27

**Decision:**

Accept (Poster)

**Comment:**

Congratulations, the paper is accepted to NeurIPS 2021!
Please reformulate your contributions in light of reviewer's qxvV comments.
Please incorporate other corrections and additions from the rebuttal/reviews.